# Bipartite Unique Neighbour Expanders via Ramanujan Graphs

**DOI:** 10.3390/e26040348

**Published:** 2024-04-20

**Authors:** Ron Asherov, Irit Dinur

**Affiliations:** Weizmann Institute, Rehovot 76100, Israel; ron.asherov@weizmann.ac.il

**Keywords:** expander graphs, pseudorandomness, unique-neighbor expansion

## Abstract

We construct an infinite family of bounded-degree bipartite unique neighbour expander graphs with arbitrarily unbalanced sides. Although weaker than the lossless expanders constructed by Capalbo et al., our construction is simpler and may be closer to being implementable in practice, due to the smaller constants. We construct these graphs by composing bipartite Ramanujan graphs with a fixed-size gadget in a way that generalises the construction of unique neighbour expanders by Alon and Capalbo. For the analysis of our construction, we prove a strong upper bound on average degrees in small induced subgraphs of bipartite Ramanujan graphs. Our bound generalises Kahale’s average degree bound to bipartite Ramanujan graphs, and may be of independent interest. Surprisingly, our bound strongly relies on the exact Ramanujan-ness of the graph and is not known to hold for nearly-Ramanujan graphs.

## 1. Introduction

An infinite family $G_{n}=\left(L_{n}⊔R_{n},E_{n}\right)$ of (c,d)-biregular graphs with $|L_{n}\left|+\right|R_{n}|\to \infty$ is called a *unique neighbour expander family* if there exists δ>0 such that, for every *n* and every set of left-side vertices $S\subseteq L_{n}$ of size $\left|S|\leq \delta \right|L_{n}|$, there exists a unique neighbour of *S* in $G_{n}$, namely, a vertex in $R_{n}$ that is connected to exactly one vertex in *S*. We only require that sets of left vertices have unique neighbours, and, arbitrarily, small right-side sets may have no unique neighbour.

Alon and Capalbo [1] constructed several explicit families of unique neighbour expanders, via an elegant composition of a Ramanujan graph and a gadget. They constructed three families of general (non-bipartite) graphs in which all small sets have unique neighbours, and one family of slightly unbalanced bipartite graphs, where small sets on the left have unique neighbours on the right. In their construction, the left side is 22/21 times bigger than the right side. The more imbalanced the graph, the harder it is for small left-hand side sets to expand into the right-hand side. Capalbo et al. [2] constructed arbitrarily unbalanced bipartite graphs that are lossless expanders, a notion strictly stronger than unique neighbour expansion. Their construction is based on a sequence of somewhat involved composition steps using randomness conductors.

Our main theorem is an efficient construction of an infinite family of bipartite unique neighbour expanders for any constant imbalance α, and any sufficiently large left-regularity degrees of a specific form.

**Theorem** **1.**
*There is a function $\hat{q}:N\times R\to N$ such that, for every integer $c_{0}>5$ and real number α>1, if $q>\hat{q}\left(c_{0},\alpha \right)$ is a prime power and $\alpha c_{0}\left(q+1\right)$ is an integer, then there is an infinite family of $\left(c_{0}\left(q+1\right),\alpha c_{0}\left(q+1\right)\right)$-biregular unique neighbour expanders. The family is constructible in polynomial time in the size of the graph.*


The theorem is proven in Section 6.2, and provides a way to compute $\hat{q}\left(c_{0},\alpha \right)$. Here are some computed values of $\hat{q}\left(c_{0},\alpha \right)$ for several values of $c_{0},\alpha$: $$\begin{array}{ccc} c_{0} & \alpha & \hat{q}\left(c_{0},\alpha \right)\\ 10 & 2 & 18907\\ 35 & 2 & 1492\\ 100 & 100 & 136051\\ 100 & 1.01 & 1135 \end{array}$$

It should be noted that $\hat{q}\left(c_{o},\alpha \right)$ increases with α, reflecting the fact that constructions with larger α (namely, more imbalanced sides) are harder to come by, and require larger degrees.

One of the prominent uses of bipartite expanders in general and bipartite unique neighbour expanders in particular, and the motivation for this work, is the construction of error-correcting codes. The work of Sipser and Spielman [3] constructs a linear error-correcting code from a bipartite unique neighbour expander, simply by looking at the adjacency matrix of the graph as a parity check matrix for the code. It is immediately obvious that the parameter δ from the unique neighbour property is a lower bound on the distance of the code.

Recently, there has also been a line of work constructing quantum LDPC codes from expanders, and a construction using lossless expansion (a strengthening of unique neighbour expansion) was introduced in [4]. Following our work, several new works [5,6,7,8] were published, which give either similar constructions or even the stronger notion of lossless expansion (thereby simplifying and reproducing the results of [2]). All of these results (including ours) only discuss the expansion of subsets of *L* (and not of *R*) and, therefore, fall short from being used in the qLDPC construction of [4].

The construction uses an infinite family of bipartite Ramanujan graphs, namely graphs whose nontrivial spectrum is contained in the spectrum of the (c,d)-biregular tree (see Section 3 for details). We constructed the unique neighbour expander family by taking a family of bipartite Ramanujan graphs and combining them with a fixed-size graph (a “gadget”), with a good unique neighbour property (small sets have unique neighbours), whose existence is shown via the probabilistic method (Lemma 6). The combination is done as follows. We first place a copy of the gadget for every right-side vertex of the Ramanujan graph. The vertex is replaced by the right side of the gadget, and its neighbours are identified with the left side of the gadget. The gadget is used to route the neighbours of each left-side vertex in the Ramanujan graph to its neighbours in the product graph.

The expansion in the product graph comes from the unique neighbour expansion of the gadget, together with low-degree vertices in induced subgraphs of the Ramanujan graph. The latter is guaranteed to exist thanks to the following (new) bound on the average degree of induced subgraphs of bipartite Ramanujan graphs, which may be of independent interest.

**Theorem** **2.***Let* G=(L⊔R,E) *be a* (c,d)*-biregular Ramanujan graph, and let* ε>0. *Then, there exists* δ>0, *which depends only on* ε,c,d, *such that, for every* S⊂L *of size* |S|≤δ|L|, *the set* N(S)⊆R *of the neighbours of S satisfies*$$\frac{c|S|}{\left|N(S)\right|}\leq 1+\left(1+\varepsilon \right)\sqrt{\frac{d-1}{c-1}}.$$

The theorem shows that every small set on the left side admits neighbours on the right side, with very few neighbours in the induced subgraph. The proof involves the recursive analysis of non-backtracking paths. Interestingly, the recursion has a nice solution only when the graph is Ramanujan. It is unclear whether this method can be extended to “nearly-Ramanujan” graphs.

We used the theorem in the following way. Given a small enough subset of left-side vertices, the theorem guarantees that it has an “almost” unique neighbour, in the sense that its degree in the induced subgraph is low (namely, it is connected to few vertices in our subset). That implies, via our construction, a small subset of vertices in a copy of the gadget; the latter has a unique neighbour in the gadget, which gives (through Lemma 7) a unique neighbour in our product graph.

Even though Ramanujan graphs are the best spectral expanders one can hope for, an efficient construction of Ramanujan graphs (be them bipartite or not) does not immediately imply that we can construct unique neighbour expanders. In the *d*-regular case, Kahale shows [9] (Thm 5.2) that there are nearly-Ramanujan graphs with an expansion of d/2 at most, which is not enough for unique neighbour expansion. In fact, recently, Kamber and Kaufman [10] proved that some Ramanujan graphs strongly fail to have unique neighbour expansion, by giving explicit constructions of arbitrarily small sets that do not admit a unique neighbour.

As mentioned previously, the graph product we defined requires a fixed-size gadget, whose proof of existence is not constructive. In principle, such a gadget could be found by an exhaustive search, since we are working in a constant-sized search space. The gadget’s size in our construction is at least cubic in *q*, so an exhaustive search is impractical, even for small values of *q*. Unfortunately, we know of no efficient construction methods for a gadget with the required parameters, and were unable to find a single working example of such a gadget. Thus, currently, our construction does not give any concrete new error-correcting codes, but rather only a scheme for the construction of such codes. It is possible that the graph sampling method presented in [11] can be used to construct fixed-size gadgets more efficiently.

The rest of this work is organised as follows. In Section 2, we survey some of the uses of unique neighbour expanders, and mention known constructions of such graphs. Section 3 provides basic definitions and results. Our main technical tool, which asserts the low induced degree in bipartite Ramanujan graphs, is stated and proven in Section 4. We prove the existence of a fixed-size gadget with good unique neighbour expansion properties in Section 5. In Section 6, we define the way we use the Ramanujan graphs and the gadget to construct bipartite unique neighbour expanders, and by that, prove Theorem 1.

## 2. Related Work

As mentioned earlier, unique neighbour expanders are motivated by the construction of error-correcting codes. In a seminal paper [3], Sipser and Spielman showed two constructions of expander codes. The first construction was obtained by simply using the adjacency matrix of a unique neighbour expander as a parity check matrix. The second was a Tanner code that is based on two components: a spectral expander graph and a constant-sized code $C_{0}$.

Our construction imitates the ideas underlying the second construction. We took a spectral (Ramanujan) expander and combined it with a small gadget graph in a way that resembles a constant-sized code $C_{0}$. Essentially, we put the local code as a gadget under the hood of our new expander construction.

If one plugs in our expander graphs into the first construction of codes from [3], these codes can be viewed as expander codes from the *second* construction, if we re-interpret the gadget as an inner code. The new point of view allows us to rely on slightly weaker features of the code, replacing odd neighbour expansion with unique neighbour expansion.

We give full details of this graph product in Section 6.1.

Unique neighbour expansion is also important for error-correcting codes with additional features. In [12,13], it is shown that codes constructed from unique neighbour expanders are weakly smooth and can be used to construct robustly testable codes. This property, in turn, is important for the composition of locally testable codes, and, more recently, for quantum LDPC codes (see [14,15]). As mentioned earlier, in the work of Hsieh and Lin [4], lossless expanders (which are a strengthening of unique neighbour expanders) are used to construct quantum LDPC codes with good rate and distance.

The uses of unique neighbour expanders are not limited to error-correcting codes: for example, such graphs may be used in the context of non-blocking networks, where it is required to connect several input–output terminals via paths in a non-intersecting fashion. Arora et al. [16] used graphs with expansion beyond the d/2 barrier to establish the existence of unique neighbours in the graph, which are useful in finding input–output paths in the online settings. Roughly speaking, when routing a set of input–output pairs, the algorithm can use all unique neighbours freely since they are guaranteed not to interfere with any other paths. Pippenger [17] used explicit constructions of spectral expanders in order to solve a similar problem, in the case where the route planning is computed locally. There, the spectral expansion of a graph was proven to imply a combinatorial expansion, in a similar way to our Theorem 2.

Another use for unique neighbour expanders is for load-balancing problems, such as the token distribution problem described in [18], and the similar pebble distribution problem, briefly discussed in [1]. In the latter, pebbles are placed arbitrarily on vertices of a graph, and need to be distributed via the edges of the graph, such that no vertex has more than one pebble. Given that the total number of pebbles is small and that the graph has the unique neighbour property, we have an efficient parallel algorithm for redistributing the pebbles.

Alon and Capalbo [1] constructed several families of unique neighbour expanders; one of them is a family of bipartite graphs whose left side is 22/21 times bigger than the right side. Similar to the construction presented in this work, each graph in the constructed family is a combination of a Ramanujan graph and a fixed graph. These graphs are not (bi)regular, but their degrees are bounded by a constant. Becker [19] used a different family of 8-regular Ramanujan graphs in order to construct a family of (non-bipartite) unique neighbour expanders, with the additional property that each graph in the family is a Cayley graph.

A slightly different approach to constructing bipartite graphs uses randomness conductors. At a high level, our construction is not much different, since it also uses a large spectral expander combined with a constant-sized gadget. However, conceptually, the difference is that the gadgets required in [2] are more involved. Randomness conductors are functions that receive a bitstring with some entropy (according to some measure of entropy) and a uniformly random bitstring, and output a bitstring with certain guarantees on its entropy. Some conductors can be constructed explicitly via a spectral method, and Capalbo et al. [2] combined them in a zig-zag-like fashion in order to construct an infinite family of *bipartite lossless expanders*, namely bipartite graphs with fixed left-regularity *c*, where small enough sets contained in the left side have at least c(1−ε) neighbours on the right side. These graphs are trivially unique neighbour expanders, since a simple counting argument shows that if a set expands by a factor of more than c/2, then it has unique neighbours.

## 3. Preliminaries

### 3.1. Expander Graphs

In this work, we deal with undirected graphs that may contain multiple edges between two vertices, but do not contain self-loops. For a graph *G* and a subset of its vertices *S*, we denote by $N_{G}\left(S\right)$ the *neighbourhood* of *S*, namely, all vertices adjacent to some vertex in *S*. When the graph in the discussion is obvious, we may omit it and write N(S). We say that *v* is a *unique neighbour* of *S* if there is a unique u∈S that is adjacent to *v*.

Let $\left(G_{n}\right)$ be a series of graphs with the number of vertices growing to infinity. There are several well-studied notions of *expansion* in graph families; we note some of them as follows:*Vertex expansion.* $\left(G_{n}\right)$ is a (δ,α)-vertex expander if, for every *n* and any subset $S\subseteq V_{G_{n}}$, if $\left|S|\leq \delta \right|V_{G_{N}}|$, we have that $|N_{G_{N}}\left(S\right)\left|\geq \alpha |S\right|$.*Edge expansion.* $\left(G_{n}\right)$ is a (δ,α)-edge expander if, for every *n* and any subset $S\subseteq V_{G_{n}}$, if $\left|S|\leq \delta \right|V_{G_{N}}|$, we have that an α-fraction, at least, of the edges with one endpoint in *S* have their other endpoint outside of *S*.*Spectral expansion.* We assume that $\left(G_{n}\right)$ are all *d*-regular, and let $A_{n}$ be the adjacency operator associated with $G_{n}$, so $A_{n}$ is indexed by vertices of $G_{n}$ and ${\left(A_{n}\right)}_{uv}$ counts how many edges there are between *u* and *v* in $G_{n}$. Let $\lambda_{1}\geq \ldots \geq \lambda_{V_{n}}$ be its spectrum. It can be seen that $\lambda_{1}=d$. Then, $\left(G_{n}\right)$ is a λ-spectral expander if for all *n* and i≠1 we have $|\lambda_{i}|\leq \lambda$.*Unique neighbour expansion.* $\left(G_{n}\right)$ is a δ-unique neighbour expander if, for every *n*, any subset $S\subseteq V_{G_{n}}$, of size $\delta |V_{G_{N}}|$ at most, has a unique neighbour.

These definitions apply to bipartite graphs $G_{n}=\left(L_{n}⊔R_{n},E_{n}\right)$ as well, with the exception that we usually consider sets contained in the left side only, and require that $L_{n}/R_{n}$ is a constant, normally greater than 1. In this case, we note that edge expansion is meaningless (since all edges leaving the left side enter the right side), and, if a bipartite graph is (c,d)-biregular, namely, if all left-side vertices have degree *c* and all right-side vertices have degree *d*, then the largest eigenvalue of the associated adjacency operator is $\sqrt{cd}$.

It can be seen that for *d*-regular graphs, the best spectral expansion we can hope for is $\alpha =2\sqrt{d-1}$. These graphs are known as *Ramanujan graphs*.

### 3.2. Bipartite Ramanujan Graphs

Ramanujan graphs have the best spectral gap [20], and their nontrivial eigenvalues are contained in the spectrum of the infinite *d*-regular tree $T_{d}$. Similarly, in the bipartite case, biregular Ramanujan graphs are defined via their relation to the infinite biregular trees: the infinite (c,d)-biregular tree $T_{c,d}$, for d>c, has the spectrum
$$\lambda \in \mathrm{spec}\left(T_{c,d}\right)\Leftrightarrow \left|\lambda \right|\in \left\{0\right\}\cup \left[\sqrt{d-1}-\sqrt{c-1},\sqrt{d-1}+\sqrt{c-1}\right]$$
(see, e.g., [21,22].) We therefore say that a finite (c,d)-biregular graph is *bipartite Ramanujan* if its nontrivial eigenvalues lie in this set. That means that every eigenvalue λ of a bipartite Ramanujan graph belongs to one of the following classes:Trivial: $\lambda =\pm \sqrt{cd}$, with eigenvectors fixed on either sides, or λ=0;$\lambda \in [\sqrt{d-1}-\sqrt{c-1},\sqrt{d-1}+\sqrt{c-1}]$ are the nontrivial positive eigenvalues;$\lambda \in [-\sqrt{c-1}-\sqrt{d-1},\sqrt{c-1}-\sqrt{d-1}]$ are the nontrivial negative eigenvalues. It should be noted that, since the graph is bipartite, λ is an eigenvalue if and only if −λ is an eigenvalue.

By an extension of the Alon–Boppana bound, given in [23], this is the best spectral gap we can hope for, at least as far as upper bounds for |λ| are concerned. We note that, unlike the *d*-regular case, we require a lower bound to |λ| too, which is essential for our proof.

While there is a vast literature on the construction of *d*-regular Ramanujan graphs (most prominently [24,25]), less is known about bipartite Ramanujan graphs. In 2014, Marcus et al. [26] proved the existence of biregular graphs with one-sided spectral gaps that resemble the Ramanujan bounds; these graphs satisfy the one-sided inequality only, namely $\left|\lambda \right|\leq \sqrt{d-1}+\sqrt{c-1}$ for every nontrivial eigenvalue λ. Gribinski et al. [27] showed a polynomial-time construction of such graphs, for every degree (d,kd) for any integers d,k. These graphs are not sufficient for our analysis, since we make explicit use of the lower bound $\left|\lambda \right|\geq \sqrt{d-1}-\sqrt{c-1}$ too.

In 2021, Brito et al. [28] proved that a random biregular graph is nearly Ramanujan with high probability. Interestingly, and unlike other works in this field, our proof strongly relies on the graph to be exactly Ramanujan, so we could not use those constructions either.

We used an explicit construction of bipartite Ramanujan graphs (with both bounds on nontrivial eigenvalues) given by Ballantine et al.:

**Theorem** **3**([29])**.**
*For every prime power q, there exists an explicit construction of a $\left(q+1,q^{3}+1\right)$-biregular Ramanujan graph.*

## 4. Vertex Expansion in Biregular Ramanujan Graphs

Our main technical tool is the following theorem showing that bipartite Ramanujan graphs exhibit excellent left-to-right expansion. As we will see shortly, this is an improvement of the expander mixing lemma. We restate the theorem for convenience.

**Theorem** **4.***Let* G=(L⊔R,E) *be a* (c,d)*-biregular Ramanujan graph, and let* ε>0. *Then, there exists* δ>0, *which depends only on* ε,c,d, *such that, for every* S⊂L *of size* |S|≤δ|L|, *the set* N(S)⊆R *of the neighbours of S satisfies*$$\frac{c|S|}{\left|N(S)\right|}\leq 1+\left(1+\varepsilon \right)\sqrt{\frac{d-1}{c-1}}.$$

We note that the quantity on the left-hand side of the inequality can be interpreted as follows. Let us observe the bipartite graph induced by taking the vertices *S* on the left and N(S) on the right. Since every left vertex has *c* outgoing edges, the total number of edges in the induced subgraph is c|S|. This means that the expression on the left-hand side of the inequality is exactly the average degree of the right side of the induced subgraph.Interestingly, the bound in this theorem is strictly stronger than what we would obtain from just applying the expander mixing lemma, which amounts to
$$\frac{c|S|}{\left|N(S)\right|}\leq \left(1+\varepsilon \right)\cdot \left(1+\frac{d-1}{c-1}+2\sqrt{\frac{d-1}{c-1}}\right).$$
See Claim 1 for details. The fact that we improve upon the expander mixing lemma is perhaps not surprising, since our analysis is based on enumerating non-backtracking paths, and not just on the magnitude of the second-largest eigenvalue. We also use lower bounds on the magnitude of all nontrivial eigenvalues, whereas the expander mixing lemma uses just upper bounds.

### 4.1. Comparison to Known Bounds

As noted above, Theorem 2 is an improvement of the bound that the expander mixing lemma gives in similar settings. For reference, we state and prove the expander mixing lemma for bipartite Ramanujan graphs. We recall that if *G* is a (c,d)-biregular Ramanujan graph, then its nontrivial eigenvalues λ satisfy $\sqrt{d-1}-\sqrt{c-1}\leq \left|\lambda \right|\leq \sqrt{d-1}+\sqrt{c-1}$. The expander mixing lemma only uses the upper bound. 

**Claim 1** (Expander Mixing Lemma for Bipartite Ramanujan Graphs)**.***Let* G=(L⊔R,E) be a (c,d)*-biregular Ramanujan graph, and let* ε>0*. Then, there exists* δ>0 *such that, for every* S⊆L *of size* |S|≤δ|L|*, the neighbourhood of S satisfies*$$\frac{c|S|}{\left|N(S)\right|}\leq \left(1+\varepsilon \right)\left(1+\frac{d-1}{c-1}+2\frac{\sqrt{d-1}}{\sqrt{c-1}}\right).$$

**Proof.** The expander mixing lemma for biregular graphs says that, for every S⊆L, T⊆R, we have
$$\left|\frac{\left|e(S,T)\right|}{\left|E\right|}-\frac{\left|S\right|}{\left|L\right|}\cdot \frac{\left|T\right|}{\left|R\right|}\right|\leq \frac{\lambda }{\sqrt{cd}}\sqrt{\frac{\left|S\right|}{\left|L\right|}\cdot \frac{\left|T\right|}{\left|R\right|}}$$
where λ is the second-largest eigenvalue of the adjacency operator associated with *G*, as defined in item 3 of Section 3.1. So, the largest eigenvalue is $\sqrt{cd}$ (we look at the spectrun of a matrix indexed by vertices of *G*, where $G_{uv}=1$ if and only if u,v are neighbours in *G*. We obtain the largest eigenvalue $\sqrt{cd}$ with an eigenvector that is $\sqrt{c}$ on the left side of *G*, and $\sqrt{d}$ on the right side of *G*.).Picking T=N(S) means all edges coming out from *S* are in the cut, namely, |e(S,T)|=c|S|. Plugging that in gives
$$\left|\frac{c|S|}{c|L|}-\frac{\left|S\right|}{\left|L\right|}\cdot \frac{\left|N(S)\right|}{\left|R\right|}\right|\leq \frac{\lambda }{\sqrt{cd}}\sqrt{\frac{\left|S\right|}{\left|L\right|}\cdot \frac{\left|N(S)\right|}{\left|R\right|}}.$$
Multiplying both sides by $\frac{\left|L\right|}{\left|S\right|}$ gives
(1)$$\left|1-\frac{\left|N(S)\right|}{\left|R\right|}\right|\leq \frac{\lambda }{\sqrt{cd}}\sqrt{\frac{\left|S\right|}{\left|L\right|}\cdot \frac{\left|N(S)\right|}{\left|R\right|}}\cdot \frac{\left|L\right|}{\left|S\right|}=\frac{\lambda }{\sqrt{cd}}\sqrt{\frac{\left|N(S)\right|}{\left|R\right|}\cdot \frac{\left|L\right|}{\left|S\right|}}=\frac{\lambda }{\sqrt{cd}}\sqrt{\frac{\left|N(S)\right|}{\left|S\right|}}\cdot \sqrt{\frac{d}{c}}=\frac{\lambda }{c}\sqrt{\frac{\left|N(S)\right|}{\left|S\right|}}$$
where we also used the fact that |E|=c|L|=d|R|.Let us assume that |S|=α|L|. Then, we can upper bound |N(S)| by
|N(S)|≤c|S|=αc|L|=αd|R|
and so, we have
$$1-\frac{\left|N(S)\right|}{\left|R\right|}\geq 1-\frac{d\alpha |R|}{\left|R\right|}=1-d\alpha .$$
We square (Equation 1) and plug in the last inequality to obtain
$${\left(1-d\alpha \right)}^{2}\leq \frac{\lambda^{2}}{c}\cdot \frac{\left|N(S)\right|}{c|S|}.$$
We recall that *G* is bipartite Ramanujan, so $\left|\lambda \right|\leq \sqrt{d-1}+\sqrt{c-1}$. We use that and rearrange as follows:
$$\begin{array}{cc} \frac{c|S|}{\left|N(S)\right|} & \leq \frac{\lambda^{2}}{c}{\left(1-d\alpha \right)}^{-2}\\ & \leq \frac{d-1+c-1+2\sqrt{d-1}\sqrt{c-1}}{c}{\left(1-d\alpha \right)}^{-2}\\ & \leq \frac{d-1+c-1+2\sqrt{d-1}\sqrt{c-1}}{c-1}{\left(1-d\alpha \right)}^{-2}\\ & =\left(1+\frac{d-1}{c-1}+2\frac{\sqrt{d-1}}{\sqrt{c-1}}\right){\left(1-d\alpha \right)}^{-2}. \end{array}$$The claim is proven by noting that there is some δ>0, such that ${\left(1-d\alpha \right)}^{-2}\leq 1+\varepsilon$ for every α<δ, namely, whenever |S|≤δ|L|. □

Kahale proved ([9], Thm 4.2) that in *d*-regular Ramanujan graphs (not necessarily bipartite), small induced subgraphs have an average degree of $\left(1+\varepsilon \right)\left(1+\sqrt{d-1}\right)$ at most. Interestingly, this result can be deduced almost immediately from Theorem 2. This is due to the following lemma, proven in Appendix A, which asserts that the *edge–vertex incidence graph* (see [3]) of a *d*-regular Ramanujan graph is a (2,d)-biregular Ramanujan graph:

**Lemma** **1.**
*Let G be a d-regular Ramanujan graph, and $G^{\prime }$ its edge–vertex incidence graph. Then, $G^{\prime }$ is a (2,d)-biregular Ramanujan graph.*


We state and prove Kahale’s bound, but we did not use it in our construction.

**Corollary** **1.**
*Let $G=(V_{G},E_{G})$ be a d-regular Ramanujan graph, and let ε>0. Then, there exists δ>0, such that, for every induced subgraph $S=(V_{S},E_{S})$ with $\delta |V_{G}|$ vertices at most, the average degree of S is, at most,*

$$\bar{d_{S}}:=\frac{2|E_{S}|}{|V_{S}|}\leq 1+\left(1+\varepsilon \right)\sqrt{d-1}.$$



**Proof.** Let $G=(V_{G},E_{G})$ be a *d*-regular Ramanujan graph and ε>0. We define $G^{\prime }=\left(L_{G^{\prime }}⊔R_{G^{\prime }},E_{G^{\prime }}\right)$ as the edge–vertex incidence graph, namely, $L_{G^{\prime }}=E_{G}$, $R_{G^{\prime }}=V_{G}$, and, for every edge e={u,v} in *G*, we have the two edges {e,u} and {e,v} in $G^{\prime }$. Since the degree of every vertex in *G* is *d*, and, since every edge has two endpoints, we have that $G^{\prime }$ is a (2,d)-biregular graph. Lemma 1 asserts that $G^{\prime }$ is Ramanujan in the bipartite sense. By Theorem 2, there exists δ>0 such that, if $T\subseteq L_{G^{\prime }}$ is of size $\delta |L_{G^{\prime }}|$ at most, then
$$\frac{2|T|}{|N_{G^{\prime }}\left(T\right)|}\leq 1+\left(1+\varepsilon \right)\sqrt{d-1}.$$A subgraph $S=(V_{S},E_{S})$ of *G* satisfies that $E_{S}$ is a subset of left-side vertices in $G^{\prime }$, $V_{S}$ is a subset of right-side vertices in $G^{\prime }$, and $V_{S}=N_{G^{\prime }}\left(E_{S}\right)$ (because if an edge is in the subgraph, then both of its endpoints are in the subgraph, and we assume that the subgraph does not contain an isolated vertex). Therefore, if $E_{S}$ is sufficiently small, namely, if $|E_{S}\left|\leq \delta \right|L_{G^{\prime }}\left|=\delta \right|E_{G}|$, then, by Theorem 2, the average degree of $N_{G^{\prime }}\left(E_{S}\right)=V_{S}$ is bounded by $1+\left(1+\varepsilon \right)\sqrt{d-1}$.We add that if we wish to find a bound for the number of vertices, we note that $|E_{S}|\leq \frac{d}{2}\left|V_{S}\right|$. So, every induced subgraph with no more than $\frac{2}{d}\delta |E_{G}\left|=\delta \right|V_{G}|$ vertices will satisfy the required average degree bound.  □

### 4.2. Proof of Theorem 2

Theorem 2 is proven by enumerating non-backtracking paths. A non-backtracking path of length *ℓ* is a sequence of edges ${\left(\left(s\left(e_{i}\right),t\left(e_{i}\right)\right)\right)}_{i=1}^{\ell }$, such that, for every *i*, $t\left(e_{i}\right)=s\left(e_{i+1}\right)$ and $s\left(e_{i}\right)\neq t\left(e_{i+1}\right)$.

For a bipartite graph *G* and a subset *S* of left-side vertices, we define $M_{\ell }\left(S\right)$ to be the number of all non-backtracking paths whose left-side vertices are all in *S*, and $M_{\ell }\left(S,G\right)$ to be the number of non-backtracking paths whose first and last left-side vertices are in *S*. Clearly, $M_{\ell }\left(S\right)\leq M_{\ell }\left(S,G\right)$, as paths of the latter type may leave S⊔N(S) (before re-entering *S* at the last step). We use a lower bound on $M_{\ell }\left(S\right)$ due to [30]:

**Lemma** **2.**
*For every undirected bipartite graph $G=(L_{G}⊔R_{G},E_{G})$ and integer ℓ, it holds that*

$$M_{\ell }\left(L_{G}\right)\geq \left|E_{G}\right|{\left(\sqrt{\left({\bar{d}}_{L}-1\right)\left({\bar{d}}_{R}-1\right)}\right)}^{\ell -1}$$

*where ${\bar{d}}_{L},{\bar{d}}_{R}$ are the average degrees of the left and right sides of G, respectively.*


We state and prove an upper bound on $M_{\ell }\left(S,G\right)$:

**Lemma** **3.**
*Let G be a (c,d)-biregular Ramanujan graph with n vertices on the left side, and S a subset of the left side. Then, for every integer ℓ*

$$M_{2\ell }\left(S,G\right)\leq \left|S\right|\left((2+\sqrt{d-1})\ell +2\right){\left(c-1\right)}^{\ell /2}{\left(d-1\right)}^{\ell /2}$$

*provided that S is small enough:*

(2)
$${|S|\left(c-1\right)}^{\ell /2}{\left(d-1\right)}^{\ell /2}\leq n.$$



Before proving the upper bound, we will show how these bounds can be combined to obtain Theorem 2.

**Proof** **of Theorem 2.**Let *ℓ* be an integer to be determined later and S⊆L a sufficiently small subset (where sufficiently small means (Equation 2)). We denote by N(S)⊆R the neighbours of *S*. The subgraph induced on S∪N(S) has c|S| edges, with all left degrees *c* and average right degree ${\bar{d}}_{R}=\frac{c|S|}{\left|N(S)\right|}$.Chaining the inequalities in Lemmas 2 and 3, we have
$$c|S|{\left(\left(c-1\right)\left({\bar{d}}_{R}-1\right)\right)}^{\frac{2\ell -1}{2}}\leq M_{2\ell }\left(S\right)\leq M_{2\ell }\left(S,V_{G}\right)\leq \left|S\right|\cdot \left((2+\sqrt{d-1})\ell +2\right)\cdot {\left(c-1\right)}^{\ell /2}{\left(d-1\right)}^{\ell /2}.$$
Simplifying, we obtain
$$\begin{array}{cc} c{\left(c-1\right)}^{\ell -\frac{1}{2}}{\left({\bar{d}}_{R}-1\right)}^{\ell -\frac{1}{2}} & \leq \left((2+\sqrt{d-1})\ell +2\right)\cdot {\left(c-1\right)}^{\ell /2}\cdot {\left(d-1\right)}^{\ell /2}\\ {\left({\bar{d}}_{R}-1\right)}^{\ell -\frac{1}{2}} & \leq \frac{\left((2+\sqrt{d-1})\ell +2\right)\sqrt{c-1}}{c}{\left(\sqrt{\frac{d-1}{c-1}}\right)}^{\ell }\\ {\bar{d}}_{R}-1 & \leq {\left(\underset{★}{\underset{︸}{\frac{\left((2+\sqrt{d-1})\ell +2\right)\sqrt{c-1}\sqrt{\bar{d}-1}}{c}}}\right)}^{1/\ell }\cdot \sqrt{\frac{d-1}{c-1}} \end{array}$$Since $\bar{d}\leq d$, we have that ★=O(ℓ), so ${★}^{1/\ell }=1+o\left(1\right)$; hence, for a fixed ε>0, there exists a constant *ℓ* (that depends only on ε,c,d), such that ${★}^{1/\ell }\leq 1+\varepsilon$; this *ℓ* determines, via inequality (Equation 2), a fixed δ, such that, whenever |S|≤δn, we have
$${\bar{d}}_{R}\leq 1+\left(1+\varepsilon \right)\sqrt{\frac{d-1}{c-1}}.$$ □

We proceed to prove Lemma 3.

For a bipartite graph $G=(L_{G}⊔R_{G},E_{G})$ and an integer *ℓ*, we define $A_{\ell }^{LL},A_{\ell }^{LR},A_{\ell }^{RL},$ and $A_{\ell }^{RR}$ as operators corresponding to non-backtracking paths of length *ℓ*, that is, $A_{\ell }^{LL}$ is an $L_{G}\times L_{G}$ matrix whose u,v entry, when $u,vs.\in L_{G}$ are left-side vertices, is the number of non-backtracking paths of length *ℓ* from *u* to *v*. The other three operators are defined similarly.

Let *M* be the operator corresponding to a single step from the right side *G* to the left side of *G*, namely, *M* has $|R_{G}|$ rows and $|L_{G}|$ columns, with $M_{uv}$ counting the number of edges between $u\in R_{G}$ and $v\in L_{G}$ in *G*. Then, the following recursive formulae hold for every integer ℓ>1:$$\begin{array}{cc} M^{⊤}A_{\ell }^{LL} & =A_{\ell +1}^{RL}+\left(d-1\right)A_{\ell -1}^{RL}\\ M^{⊤}A_{\ell }^{LR} & =A_{\ell +1}^{RR}+\left(d-1\right)A_{\ell -1}^{RR}\\ MA_{\ell }^{RL} & =A_{\ell +1}^{LL}+\left(c-1\right)A_{\ell -1}^{LL}\\ MA_{\ell }^{RR} & =A_{\ell +1}^{LR}+\left(c-1\right)A_{\ell -1}^{LR} \end{array}$$

The first formula is explained as follows. Every non-backtracking path from *R* to *L* of length ℓ+1 is composed of a non-backtracking path from *L* to *L* of length *ℓ* plus an extra step (that is the $M^{⊤}A_{\ell }^{LL}$ factor.) The opposite is true, except for paths counted in $M^{⊤}A_{\ell }^{LL}$ that do backtrack, namely those made of a non-backtracking path of length ℓ−1, and walking back and forth along the same edge. There are d−1 ways to choose that edge (since it cannot be the one that was last in the path of length ℓ−1, otherwise, it would not be counted in $M^{⊤}A_{\ell }^{LL}$), so we need to subtract $\left(d-1\right)A_{\ell -1}^{RL}$. The rest of the equations are explained in an analogue way.

Due to symmetry, we have
$${\left(A_{\ell }^{LL}\right)}^{⊤}=A_{\ell }^{LL}\;,\;{\left(A_{\ell }^{RR}\right)}^{⊤}=A_{\ell }^{RR}\;,\;{\left(A_{\ell }^{LR}\right)}^{⊤}=A_{\ell }^{RL}$$
Furthermore, since the graph is bipartite we have
$$\begin{array}{cc} A_{2\ell }^{LR} & =0\;,\;A_{2\ell }^{RL}=0\\ A_{2\ell +1}^{LL} & =0\;,\;A_{2\ell +1}^{RR}=0 \end{array}$$
for every integer *ℓ*. These equations yield a recursive formula for $A_{\ell }^{LL}$, with the following initial conditions:(3)$$\begin{array}{cc} A_{2}^{LL} & =MM^{⊤}-cI\\ A_{4}^{LL} & =MM^{⊤}A_{2}^{LL}-\left(c-1+d-1\right)A_{2}^{LL}-c\left(d-1\right)I\\ MM^{⊤}A_{\ell }^{LL} & =A_{\ell +2}^{LL}+\left(\left(c-1\right)+\left(d-1\right)\right)A_{\ell }^{LL}+\left(c-1\right)\left(d-1\right)A_{\ell -2}^{LL}\;,\;\forall \ell \geq 4 \end{array}$$

The following lemma, proven in Appendix A, suggests a way to find a non-recursive formula for $A_{2\ell }^{LL}$, given such linear recursive relations with fixed coefficients.

**Lemma** **4.**
*Let $\left(x_{n}\right)$ be a series defined via a second-order linear recurrence with fixed coefficients A,B∈C:*

$$x_{n}=Ax_{n-1}+Bx_{n-2}$$

*We assume $\lambda_{1}\neq \lambda_{2}$ are (real or complex) roots of the characteristic polynomial $\lambda^{2}-A\lambda -B$. Then, there are α,β∈C, which depend on the initial conditions $x_{0},x_{1}$, such that*

$$\;x_{n}=\alpha \lambda_{1}^{n}+\beta \lambda_{2}^{n}$$

*for every n≥0.*

*If the characteristic polynomial has a single root λ of multiplicity 2, then there are α,β∈C, such that*

$$x_{n}=\alpha \lambda^{n}+\beta n\lambda^{n}$$

*for every n≥0.*


We use the lemma to bound the eigenvalues of $A_{2\ell }^{LL}$ given bounds on the spectrum of the biregular graph.

**Lemma** **5.**
*Let G be a (c,d)-biregular graph. Then, there is a sequence of polynomials with integer coefficients $\left(p_{\ell }\left(x\right)\right)$ such that, for every eigenvalue v of G with eigenvalue λ, $p_{\ell }\left(\lambda^{2}\right)$ is an eigenvalue of $A_{2\ell }^{LL}$, and, moreover, for every λ∈R, if*

(4)
$$\left|\lambda \right|\in \left\{0\right\}\cup \left[\sqrt{d-1}-\sqrt{c-1},\sqrt{d-1}+\sqrt{c-1}\right]$$

*then*

(5)
$$|p_{\ell }\left(\lambda^{2}\right)|\leq \left(2+\sqrt{d-1}\right)\ell {\left(c-1\right)}^{\ell /2}{\left(d-1\right)}^{\ell /2}.$$



**Proof.** The recursive formulae proven above (Equation 3) suggest that there is a series of polynomials $p_{n}\left(x\right)$ with integer coefficients such that $A_{2n}^{LL}=p_{n}\left(MM^{⊤}\right)$. It should be noted that the graph’s adjacency matrix is
$$A_{G}=\left[\begin{array}{cc} 0 & M\\ M^{⊤} & 0 \end{array}\right]$$
Furthermore, if (λ,v) is an eigenpair of *G*, then $\left(\lambda^{2},v\right)$ is an eigenpair of
$$A_{G}^{2}=\left[\begin{array}{cc} MM^{⊤} & 0\\ 0 & M^{⊤}M \end{array}\right].$$
This shows that $p_{\ell }\left(\lambda^{2}\right)$ is an eigenvalue of $A_{2\ell }^{LL}$ whenever λ is an eigenvalue of *G*. The converse is also true.The formulae (Equation 3) can be transformed so as to convey that $p_{n}\left(x\right)$ satisfies the following equations:
$$\begin{array}{c} p_{1}\left(x\right)=x-c\;,\;p_{2}\left(x\right)=x^{2}+\left(2-2c-d\right)x+c\left(c-1\right)\\ xp_{n}\left(x\right)=p_{n+1}\left(x\right)+\left(c-1+d-1\right)p_{n}\left(x\right)+\left(c-1\right)\left(d-1\right)p_{n-1}\left(x\right) \end{array}$$
for all n>1. Setting n=1 gives an equation involving $p_{0}\left(x\right),p_{1}\left(x\right),$ and $p_{2}\left(x\right)$. We can solve this equation for $p_{0}\left(x\right)$ and obtain a simpler description of the initial conditions:
(6)$$\begin{array}{c} p_{0}\left(x\right)=\frac{c}{c-1}\;,\;p_{1}\left(x\right)=x-c \end{array}$$
(7)$$\begin{array}{c} xp_{n}\left(x\right)=p_{n+1}\left(x\right)+\left(c-1+d-1\right)p_{n}\left(x\right)+\left(c-1\right)\left(d-1\right)p_{n-1}\left(x\right) \end{array}$$
for all n>0.We fix some *t* that satisfies (Equation 4), namely,
$$\left|t\right|\in \left\{0\right\}\cup \left[\sqrt{d-1}-\sqrt{c-1},\sqrt{d-1}+\sqrt{c-1}\right].$$
We first deal with the case where $\left|t|\in (\sqrt{d-1}-\sqrt{c-1},\sqrt{d-1}+\sqrt{c-1}\right)$, and, later, we will consider the edge cases where *t* is one of the endpoints of the segment or 0. Let us write $x=t^{2}$. We have that, for this fixed *x*, the series ${\left(p_{n}\left(x\right)\right)}_{n}$ satisfies a second-order linear recurrence with fixed coefficients. Using Lemma 4, we conclude that there are functions $\alpha \left(x\right),\lambda_{1}\left(x\right),\beta \left(x\right),$ and $\lambda_{2}\left(x\right)$ that depend only on x,c, and *d*, such that
(8)$$p_{n}\left(x\right)=\alpha \left(x\right){\left(\lambda_{1}\left(x\right)\right)}^{n}+\beta \left(x\right){\left(\lambda_{2}\left(x\right)\right)}^{n}$$
for every *n*.In order to find $\lambda_{1},\lambda_{2}$, we solve for λ the characteristic polynomial, namely, the following quadratic equation derived from (7):
$$x\lambda =\lambda^{2}+\left(c-1+d-1\right)\lambda +\left(c-1\right)\left(d-1\right)$$
to obtain
$$\lambda_{1,2}\left(x\right)=\frac{x-\left(c-1\right)-\left(d-1\right)\pm \sqrt{\Delta (x)}}{2}$$
where
(9)$$\Delta \left(x\right)=x^{2}-2x\left(\left(c-1\right)+\left(d-1\right)\right)+{\left(c-d\right)}^{2}.$$
Using the initial values for $p_{0}\left(x\right),p_{1}\left(x\right)$ from (Equation 6), and plugging back into (Equation 8), we obtain the equations
$$\begin{array}{cc} \frac{c}{c-1} & =\alpha \left(x\right){\left(\lambda_{1}\left(x\right)\right)}^{0}+\beta \left(x\right){\left(\lambda_{2}\left(x\right)\right)}^{0}=\alpha \left(x\right)+\beta \left(x\right)\\ x-c & =\alpha \left(x\right){\left(\lambda_{1}\left(x\right)\right)}^{1}+\beta \left(x\right){\left(\lambda_{2}\left(x\right)\right)}^{1}=\alpha \left(x\right)\lambda_{1}\left(x\right)+\beta \left(x\right)\lambda_{2}\left(x\right) \end{array}$$
whose solution is
$$\begin{array}{cc} \alpha (x) & =\frac{\left(c-1\right)x-{\left(c-1\right)}^{2}-\left(c-1\right)+\left(c-1\right)\left(d-1\right)+\left(c-1\right)\sqrt{\Delta (x)}-x+d-1+\sqrt{\Delta (x)}}{2\left(c-1\right)\sqrt{\Delta (x)}}\\ \beta (x) & =\frac{c}{c-1}-\alpha \left(x\right). \end{array}$$We check when Δ(x)=0 by solving (Equation 9) for *x*:
$$\begin{array}{cc} x_{1,2} & =\frac{2\left(\left(c-1\right)+\left(d-1\right)\right)\pm \sqrt{4{\left(c-1+d-1\right)}^{2}-4{\left(c-d\right)}^{2}}}{2}\\ & =\left(c-1+d-1\right)\pm \sqrt{{\left(c+d\right)}^{2}-4\left(c+d\right)+4-{\left(c-d\right)}^{2}}\\ & =\left(c-1+d-1\right)\pm \sqrt{c^{2}+2cd+d^{2}-4c-4d+4-c^{2}+2cd-d^{2}}\\ & =\left(c-1+d-1\right)\pm \sqrt{4cd-4c-4d+4}\\ & =\left(c-1+d-1\right)\pm 2\sqrt{c-1}\sqrt{d-1}\\ & ={\left(\sqrt{d-1}\pm \sqrt{c-1}\right)}^{2} \end{array}$$We see that Δ(x) is quadratic in *x* and has roots at ${\left(\sqrt{d-1}\pm \sqrt{c-1}\right)}^{2}$. This gives a nice factorization of Δ(x):
$$\begin{array}{cc} \Delta (x) & =x^{2}-2x\left(\left(c-1\right)+\left(d-1\right)\right)+{\left(c-d\right)}^{2}\\ & =\left(x-{\left(\sqrt{d-1}+\sqrt{c-1}\right)}^{2}\right)\left(x-{\left(\sqrt{d-1}-\sqrt{c-1}\right)}^{2}\right) \end{array}$$We recall that, for the *x* we fixed, we have $\sqrt{x}=t\in \left(\sqrt{d-1}-\sqrt{c-1},\sqrt{d-1}+\sqrt{c-1}\right)$, so the first term in the product is negative and the second term is positive, so Δ<0, and $\lambda_{1,2}$ are complex numbers (conjugate to one another), with magnitude
(10)$$\begin{array}{cc} |\lambda_{1,2}{|}^{2} & =\frac{{\left(x-\left(c-1\right)-\left(d-1\right)\right)}^{2}-\Delta \left(x\right)}{4}\\ & =\frac{x^{2}-2x\left(\left(c-1\right)+\left(d-1\right)\right)+{\left(c-1+d-1\right)}^{2}-\left(x^{2}-2x\left(\left(c-1\right)+\left(d-1\right)\right)+{\left(c-d\right)}^{2}\right)}{4}\\ & =\frac{{\left(c+d-2\right)}^{2}-{\left(c-d\right)}^{2}}{4}=\left(c-1\right)\left(d-1\right) \end{array}$$A very similar calculation shows that α,β are conjugates with magnitude
$${\left|\alpha \right|}^{2}={\left|\beta \right|}^{2}=\frac{x(x-cd)}{\Delta (x)\cdot (c-1)}$$This finishes the proof for all such *x*s:
$$\begin{array}{cc} |p_{\ell }\left(x\right)| & =|\alpha \left(x\right)\lambda_{1}^{\ell }+\beta \left(x\right)\lambda_{2}^{\ell }\left|\leq \right|\alpha \left(x\right)\lambda_{1}^{\ell }\left|+\right|\beta \left(x\right)\lambda_{2}^{\ell }|\\ & =\left|\alpha (x)|\right|\lambda_{1}{|}^{\ell }+\left|\beta (x)|\right|\lambda_{2}{|}^{\ell }\\ & =2\sqrt{\frac{x(x-cd)}{\Delta (x)\cdot (c-1)}}{\left(c-1\right)}^{\ell /2}{\left(d-1\right)}^{\ell /2} \end{array}$$We keep in mind that *x* is fixed, so the expression is smaller than $\left(2+\sqrt{d-1}\right)\cdot \ell \cdot {\left(c-1\right)}^{\ell /2}{\left(d-1\right)}^{\ell /2}$ for large enough *ℓ*.We are left with the cases $x=t^{2}$ for $t=0,\sqrt{d-1}\pm \sqrt{c-1}$:
t=0. We use the same methods and find that the characteristic polynomial is
$$\lambda^{2}+\left(c-1+d-1\right)\lambda +\left(c-1\right)\left(d-1\right)$$
whose roots are
$$\lambda_{1}=-\left(c-1\right)\;,\;\lambda_{2}=-\left(d-1\right).$$Using the initial conditions ($p_{0}\left(0\right)=c/\left(c-1\right)$ and $p_{1}\left(0\right)=-c$), we obtain
$$\alpha \left(0\right)=\frac{c}{c-1}\;,\;\beta \left(0\right)=0$$
and, using the fact that c<d, we obtain
$$\begin{array}{cc} |p_{\ell }\left(0\right)| & =|\alpha \left(0\right)\lambda_{1}^{\ell }+\beta \left(0\right)\lambda_{2}^{\ell }|\\ & =\frac{c}{c-1}{\left(c-1\right)}^{\ell }\\ & <2l{\left(c-1\right)}^{\ell /2}{\left(c-1\right)}^{\ell /2}\\ & <2l{\left(c-1\right)}^{\ell /2}{\left(d-1\right)}^{\ell /2}. \end{array}$$$t=\sqrt{d-1}+\sqrt{c-1}$. Then, $x=t^{2}={\left(\sqrt{d-1}+\sqrt{c-1}\right)}^{2}=d-1+c-1+2\sqrt{d-1}\sqrt{c-1}$, and the characteristic polynomial has a single root of multiplicity 2, namely,
$$\lambda =\frac{x-(c-1)-(d-1)}{2}=\sqrt{d-1}\sqrt{c-1}.$$The solution, therefore, takes the form
$$p_{n}\left(x\right)=\left(\alpha \left(x\right)+n\beta \left(x\right)\right){\left(c-1\right)}^{n/2}{\left(d-1\right)}^{n/2}.$$Using the initial values, we obtain
$$\alpha \left(x\right)=\frac{c}{c-1}\;,\;\beta \left(x\right)=\frac{x-c}{\sqrt{d-1}\sqrt{c-1}}-\frac{c}{c-1}=2+\frac{d-2}{\sqrt{d-1}\sqrt{c-1}}-\frac{c}{c-1}.$$$1<\frac{c}{c-1}\leq 2$ so $\beta \left(x\right)\leq \sqrt{d-1}+1$, and, in total, we obtain
$$\begin{array}{cc} |p_{\ell }\left(x\right)| & ={|\alpha (x)+\ell \beta (x)|\left(c-1\right)}^{\ell /2}{\left(d-1\right)}^{\ell /2}\\ & \leq \left(\left|\frac{1}{\ell }\cdot \frac{c}{c-1}\right|+\left|\beta (x)\right|\right)\ell {\left(c-1\right)}^{\ell /2}{\left(d-1\right)}^{\ell /2}\\ & \leq \left(2+\sqrt{d-1}\right)\ell {\left(c-1\right)}^{\ell /2}{\left(d-1\right)}^{\ell /2} \end{array}$$
for sufficiently large *ℓ*.$t=\sqrt{d-1}-\sqrt{c-1}$. We obtain $x=t^{2}=d-1+c-1-2\sqrt{d-1}\sqrt{c-1}$, and the rest follows the same calculations as in the previous case. □

Bounds on the spectrum of $A_{2\ell }^{LL}$ give bounds on the number of non-backtracking paths completely contained in a small set, hence, give Lemma 3.

**Proof** **of Lemma 3.**We recall that $M_{2\ell }\left(S,G\right)$ counts the number of non-backtracking paths of length 2ℓ that start and end in *S*, so, by the definition of the $A_{n}^{LL}$ operator, we have $M_{2\ell }\left(S,G\right)=\langle A_{2\ell }^{LL}1_{S},1_{S}\rangle$.We note that $A_{2\ell }^{LL}1_{L}=c{\left(c-1\right)}^{\ell -1}{\left(d-1\right)}^{\ell }1_{L}$, because every non-backtracking path starting at a given vertex is made by picking the first left-to-right edge (we have *c* such edges to pick from), and then alternating between picking any of the *d* or *c* edges adjacent to the current vertex, except for the edge we picked to get to it.We write $1_{S}=\frac{\left|S\right|}{n}1_{L}+r$, with $r⊥1_{L}$, and ${∥r∥}_{2}^{2}\leq {∥1_{S}∥}_{2}^{2}=\left|S\right|$. Since the graph is Ramanujan, the nontrivial eigenvalues in its spectral decomposition have their absolute value in the set $\left\{0\right\}\cup \left[\sqrt{d-1}-\sqrt{c-1},\sqrt{d-1}+\sqrt{c-1}\right]$. We only care about the nontrivial eigenvalues because $r⊥1_{L}$; hence, in the writing of *r* in the orthogonal basis made of eigenvectors, only eigenvectors with nontrivial eigenvalues appear. We decompose $r=\sum \alpha_{i}v_{i}$, and, using the notation in Lemma 5, we obtain
$$A_{2\ell }^{LL}r=A_{2\ell }^{LL}\sum_{i}\alpha_{i}v_{i}=\sum_{i}\alpha_{i}A_{2\ell }^{LL}v_{i}=\sum_{i}p_{\ell }\left(\lambda_{i}^{2}\right)\alpha_{i}v_{i},$$
and, with the result of that lemma, we have
$$\langle A_{2\ell }^{LL}r,r\rangle =\sum_{i}p_{\ell }\left(\lambda_{i}^{2}\right)\alpha_{i}^{2}\leq \left(2+\sqrt{d-1}\right)\ell {\left(c-1\right)}^{\ell /2}{\left(d-1\right)}^{\ell /2}\cdot {∥r∥}_{2}^{2}.$$
We combine everything to obtain
$$\begin{array}{cc} M_{2\ell }\left(S,G\right) & =\langle A_{2\ell }^{LL}1_{S},1_{S}\rangle =\langle A_{2\ell }^{LL}\left(\frac{\left|S\right|}{n}1_{L}+r\right),\frac{\left|S\right|}{n}1_{L}+r\rangle \\ & =\frac{{\left|S\right|}^{2}}{n^{2}}\langle A_{2\ell }^{LL}1_{L},1_{L}\rangle +\langle A_{2\ell }^{LL}r,r\rangle \\ & =\frac{{\left|S\right|}^{2}}{n^{2}}\cdot c{\left(c-1\right)}^{\ell -1}{\left(d-1\right)}^{\ell }\langle 1_{L},1_{L}\rangle +\langle A_{2\ell }^{LL}r,r\rangle \\ & \leq \frac{{\left|S\right|}^{2}}{n}c{\left(c-1\right)}^{\ell -1}{\left(d-1\right)}^{\ell }+\left(2+\sqrt{d-1}\right)\ell {\left(c-1\right)}^{\ell /2}{\left(d-1\right)}^{\ell /2}{∥r∥}_{2}^{2}\\ & \leq \left|S\right|\left(\frac{\left|S\right|\cdot c\cdot {\left(c-1\right)}^{\ell /2}{\left(d-1\right)}^{\ell /2}}{n(c-1)}+\left(2+\sqrt{d-1}\right)\ell \right){\left(c-1\right)}^{\ell /2}{\left(d-1\right)}^{\ell /2}\\ & \leq \left|S\right|\left(\frac{c}{c-1}+\left(2+\sqrt{d-1}\right)\ell \right){\left(c-1\right)}^{\ell /2}{\left(d-1\right)}^{\ell /2}\\ & \leq \left|S\right|\left((2+\sqrt{d-1})\ell +2\right){\left(c-1\right)}^{\ell /2}{\left(d-1\right)}^{\ell /2} \end{array}$$ □

## 5. Random Gadget

In this section, we prove the existence of bipartite graphs, such that every small set of left-side vertices has a unique neighbour on the right side. We drew a random biregular graph from a similar distribution as in [31], and used techniques similar to [32] (Thm 4.4).

**Lemma** **6.**
*For every integer L,R,c, and d with Lc=Rd, L>R, c>3, if k is an integer that satisfies the inequality*

$$k^{\frac{c-3}{2}}\leq \frac{1}{2Le}\cdot {\left(\frac{R}{3ec}\right)}^{\frac{c-1}{2}}$$

*then there is a (c,d)-biregular graph with sides [L] and [R], such that every set of left vertices of size k at most has a unique neighbour.*


We drew a random (c,d)-biregular graph in the following way: we fixed *L* vertices on the left side and *R* vertices on the right side (cL=dR), wrote *c* copies of each left-side vertex and *d* copies of each right-side vertex, and connected them via a uniformly random matching. That is, we picked a uniformly random permutation π:L×[c]→R×[d], and, for every u∈L,vs.∈R,i∈[c],j∈[d], if π(u,i)=(v,j), then added (u,v) as an edge. It should be noted that we allowed multiple edges between two vertices (if there were several i,j satisfying π(u,i)=(v,j)).

Let *G* be a random bipartite graph with *L* vertices on the left side and *R* vertices on the right side drawn from said distribution. Let *A* be a subset of left vertices of size *k*. We note that, if *A* expands by at least (c+1)/2, then, by a simple counting argument, *A* has a unique neighbour. It is therefore sufficient to find the probability that *A* expands by at least (c+1)/2.

Let us fix an arbitrary ordering of the ck edges leaving *A*, and denote it $e_{1},\ldots ,e_{ck}$. We say that $e_{i}$ is a *repeat* if it touches a previously covered vertex, that is, if its right endpoint is contained in the set of right endpoints of the set $e_{1},\ldots ,e_{i-1}$. We note that if *A* does *not* expand by at least (c+1)/2, then, again, by a simple counting argument, there are at least (c−1)k/2 repeats. This is because the number of repeats and the size of the set of the neighbours of *A* add up to the number of edges leaving *A*, namely ck.

We note that for every *i*, $e_{i}$ is a repeat if it touches one of i−1 or less previously covered vertices. This means that $\mathrm{Pr}[e_{i}\;\mathrm{is}\;a\;\mathrm{repeat}]\leq \frac{i-1}{R}<\frac{ck}{R}$. Moreover, if we condition on the event that some of the first i−1 edges are also repeats, then the probability that $e_{i}$ is a repeat may only decrease, since it means that there are less “forbidden” endpoints. We conclude that, for every set of *l* edges,
$$\mathrm{Pr}[e_{i_{1}},\ldots ,e_{i_{l}}\;\mathrm{are}\;\mathrm{repeats}]=\prod_{j=1}^{l}\mathrm{Pr}[e_{i_{j}}\;\mathrm{is}\;a\;\mathrm{repeat}|e_{i_{1}},\ldots e_{i_{j-1}}\;\mathrm{are}\;\mathrm{repeats}]<{\left(\frac{ck}{R}\right)}^{l}.$$

If *A* expands too little, then there are many repeats. We can use it to bound the probability that *A* has no unqiue neighbour: Pr[Ahasnouniqueneighbour]≤Pr[Aexpandsby<(c+1)/2]≤Pr[thereareatleast(c−1)k/2repeats]≤∑i1,…,i(c−1)k/2∈ck(c−1)k/2Pr[{ei1,…,ei(c−1)k/2}arerepeats]<ckc−12k·ckRc−12k

Furthermore, by a union bound over the possible choices of *A*,
(11)Pr[∃“bad”Aofsizek]≤Lk·Pr[Aexpandsby<(c+1)/2]≤Lk·ckc−12k·ckRc−12k≤Lekk·ckec−12kc−12k·ckRc−12k=Lek·2cec−1·ckRc−12k≤Lek·3eckRc−12k
where the last inequality follows from assuming that c≥3, so $\frac{2c}{c-1}\leq 3$.

We are now ready to prove Lemma 6.

**Proof** **of Lemma 6.**Let us draw a (c,d)-biregular graph G=([L]⊔[R],E) from the distribution described above. Let *k* be an integer satsifying (6). Using a union bound and the inequality in (Equation 11), we have (where probability is taken over the choice of *G*)
$$\begin{array}{cc} \mathrm{Pr}[\exists \;“\mathrm{bad}”\;A\subseteq [L]\;\mathrm{of}\;\mathrm{size}\;\leq k] & =\sum_{a=1}^{k}\mathrm{Pr}[\exists \;“\mathrm{bad}”\;A\subseteq [L]\;\mathrm{of}\;\mathrm{size}\;a]\\ & \leq \sum_{a=1}^{k}{\left(\frac{Le}{a}\cdot {\left(\frac{3eca}{R}\right)}^{\frac{c-1}{2}}\right)}^{a}\\ & <\sum_{a=1}^{\infty }{\left(\frac{Le}{k}\cdot {\left(\frac{3eck}{R}\right)}^{\frac{c-1}{2}}\right)}^{a}\\ & =\sum_{a=1}^{\infty }{\left(k^{\frac{c-1}{3}}\cdot Le\cdot {\left(\frac{3ec}{R}\right)}^{\frac{c-1}{2}}\right)}^{a}\\ & \leq \sum_{a=1}^{\infty }{\left(\frac{1}{2}\right)}^{a}<1. \end{array}$$We see that with strictly positive probability, a random graph has no “bad” subsets of size ≤k; hence, there exists a graph with the desired unique neighbour property. □

## 6. Construction

### 6.1. Routed Product Definition

Let us begin with a brief coding theory motivation. An error-correcting code is often given via an m×n parity check matrix *H*, so that $C=\mathrm{Ker}H\subseteq {\left\{0,1\right\}}^{n}$. The matrix *H* can be visualised as a bipartite graph, called the *parity check graph*, with *n* left and *m* right vertices, and an edge i∼j whenever H(j,i)≠0. A Tanner code is defined given a bipartite graph *B* and a base code $C_{0}=\mathrm{Ker}H_{0}$ [33]. One way to view the routed product is through the point of view of codes. We consider the parity check graph $B_{0}$ of $H_{0}$ and define the *routed product* of *B* and $B_{0}$ to be simply the parity check graph of the Tanner code $C(B,C_{0})$.

Here is a more detailed and combinatorial definition of the routed product without mention of codes: let G=(L⊔R,E) be a (c,d)-biregular graph and $G_{0}=\left(L_{0}⊔R_{0},E_{0}\right)$ a $\left(c_{0},d_{0}\right)$-biregular graph.

We think of *G* as a big graph (in practice, an infinite family of Ramanujan graphs), and $G_{0}$ as a fixed-size graph (gadget). We assume that $|L_{0}|=d$, and think of the edges of *G* as a function E:R×[d]→L which maps a right-side vertex *v* and an index *i* to the *i*th neighbour of *v* in *G*.

We can define the *routed product* graph $G^{\prime }=G\circ G_{0}$ as the bipartite graph whose left side is *L*, right side is the Cartesian product $R\times R_{0}$, and the set of edges is
$$E^{\prime }=\left\{\left(E(v,i),(v,j)\right):\;vs.\in R,i\in \left[d\right],j\in \left[R_{0}\right],\left(i,j\right)\in E_{0}\right\}.$$
That is, we write $R_{0}$ copies of each vertex in *R*, and every right-side vertex *v* in the big graph *G* and an edge (i,j) in the small gadget $G_{0}$ gives an edge between the *i*th neighbour of *v* in *G*, and the $j^{\mathrm{th}}$ vertex of the copy of $G_{0}$ assigned to *v* in $G^{\prime }$. Otherwise put, we use $G_{0}$ to route every edge of the big graph *G* to $c_{0}$ edges in the product graph $G^{\prime }$.

More precisely, for every v∈R, the bipartite subgraph of $G^{\prime }$, whose left side is $N_{G}\left(v\right)$ and right side is (v,·), is isomorphic to $G_{0}$. This means that, roughly speaking, unique neighbours are inherited from the small graph to the product graph:

**Lemma** **7.**
*Let S⊆L, $v\in N_{G}\left(S\right)$. We define $S^{\prime }=\left\{i:E\left(v,i\right)\in S\right\}\subseteq \left[d\right]$ as the indexed neighbours of v in S. If $S^{\prime }$, as a set of vertices in the gadget $G_{0}$, has a unique neighbour $j\in R_{0}$ in $G_{0}$, then (v,j) is a unique neighbour of S in the product graph $G^{\prime }$.*


The proof is immediate while observing Figure 1, but, for the sake of completion, it is given in Appendix A.

### 6.2. Proof of Theorem 1

Let *q* be a prime power, $c_{0}$ an integer, and α>1. We assume that $\alpha c_{0}\left(q+1\right)$ is an integer. We construct an infinite family of $\left(c_{0}\left(q+1\right),\alpha c_{0}\left(q+1\right)\right)$-biregular graphs with the unique neighbour property under some assumptions specified below.

We denote c=q+1 and $d=q^{3}+1$. By Theorem 3, there is an efficient construction of an infinite family of (c,d)-biregular Ramanujan graphs $\left(G_{n}\right)$. Let $G_{0}=\left(L_{0}⊔R_{0},E_{0}\right)$ be a gadget: a $c_{0}$-left-regular bipartite graph with $|L_{0}|=d=q^{3}+1$ vertices on the left side and $R_{0}$ vertices on the right side, such that every left-side set of sufficiently small size admits a unique neighbour on the right side, where “sufficiently small” means the bound given in Lemma 6. For the constructed graph to have the left side α times bigger than the right side, we set $R_{0}=\frac{d}{\alpha c}=\frac{q^{3}+1}{\alpha (q+1)}$.

We define $G_{n}^{\prime }=G_{n}\circ G_{0}$ as the routed product of $G_{n}$ and $G_{0}$. For the rest of this (short) proof, let us suppress *n* from the notation, for convenience.

Let $\varepsilon <\frac{1}{q}$. By Theorem 2, there exists δ>0, such that, for every S⊆L of size δ|L| at most, the “average right degree” ${\bar{d}}_{S}$, namely, the average of the degrees of vertices in $N_{G}\left(S\right)$ in the induced subgraph $S⊔N_{G}\left(S\right)$, is bounded:$${\bar{d}}_{S}:=\frac{c|S|}{|N_{G}\left(S\right)|}\leq 1+\left(1+\varepsilon \right)\sqrt{\frac{d-1}{c-1}}.$$
We show that such *S* has a unique neighbour in $G^{\prime }$.

We note that $\frac{d-1}{c-1}=q^{2}$, so, since $\varepsilon <\frac{1}{q}$, we have a vertex v∈R of “degree” q+1 at most in *G*, that is, the set $S^{\prime }\subseteq \left[d\right]$ of *v*’s neighbours in *S* is of size q+1 at most. By Lemma 7, if $S^{\prime }$, as a set of left-side vertices in $G_{0}$, has a unique neighbour *j* in $G_{0}$, then our original set *S* has a unique neighbour (v,j) in *G*.

The parameters now need to be chosen in a way that all left-side sets of size q+1 at most have a unique neighbour in $G_{0}$. By Lemma 6, we need to have
(12)$${\left(q+1\right)}^{\frac{c_{0}-3}{2}}\leq \frac{1}{2(q^{3}+1)e}\cdot {\left(\frac{\frac{q^{3}+1}{\alpha (q+1)}}{3ec_{0}}\right)}^{\frac{c_{0}-1}{2}}.$$
The LHS is $O(q^{\frac{c_{0}-3}{2}})$ and RHS is $\Theta (q^{c_{0}-4})$, so, if $c_{0}>5$, then, for sufficiently large *q*, the construction gives a unique neighbour expander. That is, there exists some $\hat{q}\left(c_{0},\alpha \right)$, such that, if $q>\hat{q}$, then (Equation 12) holds; hence, we constructed a bipartite unique neighbour expander, as promised in Theorem 1.

## 7. Future Work

The main pitfall of our approach is the non-constructive nature of the gadget. Theoretically, since the gadget has a constant size, this is no issue. However, exhaustive search is impractical, even for small values of *q*. This is because the gadget’s size is cubic in *q*, so the search space is of exponential size in $q^{3}$. A natural question would be whether it is possible to construct such a gadget in an efficient way, since that would lead to the whole unique neighbour expander family to be constructible in practice. For the bipartite Ramanujan family chosen in our work (the one by Ballantine et al. [29]), we ask the following:

**Question** **1.***For which prime power q and real number* α≥1 *can one efficiently construct a biregular graph with left side* $q^{3}+1$ *and right side* $\frac{q^{3}+1}{\alpha (q+1)}$*, such that every left-side set of size* q+1 *at most has a unique neighbour?*

We note that the fixed-size graph given in [1] (Lemma 4.3) is a good gadget (for α=22/21 and the edge–vertex incidence graphs of a 44-regular Ramanujan graph family), and, indeed these graphs can be used to construct bipartite unique neighbour expanders.

Since we proved that a random gadget is, with non-negligble probability, good for our construction, it may be interesting to construct such a gadget by simply drawing random gadgets and testing whether they are good. Since drawing is simple, we are left with the task of testing. We therefore ask

**Question** **2.**
*Given a bipartite graph, can one efficiently find the smallest nonempty set of left-side vertices that has no unique neighbours?*


We currently know of no better way than just enumerating all left-side sets, which is exponential in the size of the graph, hence impractical. We refer to [11] for an interesting approach to testing the expansion of random graphs.

The methods presented in this work are not limited to the $\left(q+1,q^{3}+1\right)$-biregular Ramanujan family. We can therefore ask the question the other way around—find a gadget (by sampling or any other way)—and see whether we can efficiently construct a bipartite Ramanujan family that will make it work, i.e., that would allow us to rewrite the proof of Theorem 1. This emphasises the well-known natural question of constructing Ramanujan graphs with arbitrary degrees, specifically in the bipartite and biregular setting,

**Question** **3.**
*For which integers c<d can one efficiently construct an infinite family of (c,d)-biregular Ramanujan graphs?*


We note that our construction is far from “right-side unique neighbour expansion,” as the complete right side of a single gadget is a constant-sized set with no unique neighbours on the left. We wonder whether it is possible to construct a bipartite graph where *all* small size sets (be they contained in either side, or both) have unique neighbours.

## Figures and Tables

**Figure 1 entropy-26-00348-f001:**
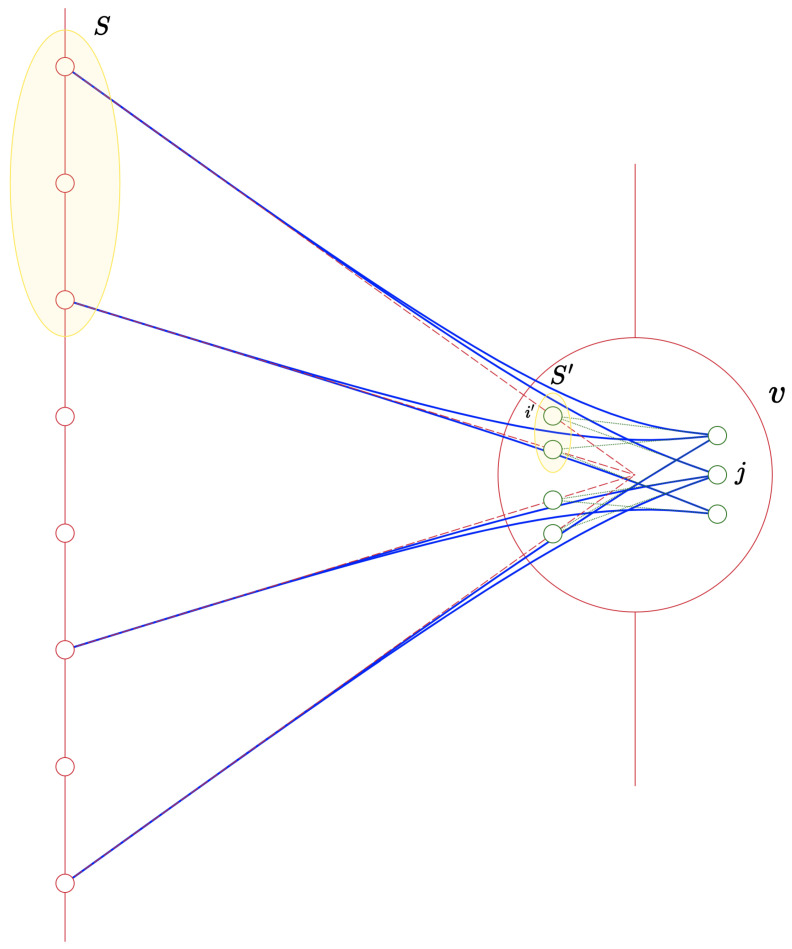
An example of a bipartite graph *G* (dashed, red), a small gadget $G_{0}$ (dotted, green), and the routed product $G^{\prime }=G\circ G_{0}$ (solid, blue). The set S⊆L has a neighbour v∈R, and so *S* is associated with a set $S^{\prime }$ of left-side vertices of the copy of $G_{0}$ associated with *v*. Since $\left(i^{\prime },j\right)$ is the only edge connecting *j* to $S^{\prime }$ in $G_{0}$, we have that (v,j) is a unique neighbour of *S* in $G^{\prime }$.

## Data Availability

Data are contained within the article.

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
