# Peer review of "Bipartite Unique Neighbour Expanders via Ramanujan Graphs"

_entropy, 2024, doi:10.3390/e26040348_

Round 1

Reviewer 1 Report

Comments and Suggestions for Authors

Paper summary:

The paper's topic is expansion properties of bi-partite graphs constructed from Ramanujan graphs, in particular proving a property called "unique-neighbor expansion". Unique-neighbor expansion is a desirable property in several applications: if every set of left-vertices (of some size) has a vertex in its right-neighborhood that connects to only one vertex in this set, then intuitively the connectivity of the graph is good. The main contribution of the work is a construction that composes a Ramanujan bi-partite graph (from a known infinite family) with a proposed constant-size gadget, such that bi-partite graphs with the above property are obtained for a wide choice of parameters. The properties of the proposed gadget are proved using the probabilistic method, that is, it is proven to exist, but not found explicitly. As pointed by the authors, this method of construction by composition of bi-partite graphs is not dissimilar from Tanner's classical graph-code construction, hence the relevance of this work to the area of coding theory. The results are proven using two technical ingredients: first a new upper bound is derived on the average induced right-degree in the Ramanujan graph, and then this bound is used by the gadget construction such that the unique-neighbor property is guaranteed (in the gadget, as well as in the composed graph).

Paper evaluation:

The paper advances the theoretical state-of-the-art of an important combinatorial object, and the results are presented in a rigorous and mostly clear way (both the novel results and the background). In that sense, it has high merit for publication. On the flip side, the paper leaves an average coding-theory reader with many questions about the motivation and meaning of the results (some examples below). Hence, to address this audience of the journal, I recommend the authors to add a section that answers at least part of these questions. Sample questions:

*) what new code parameters are implied by the results?

*) how prior expander-code constructions need to be adapted to use the new graphs.

*) can you show fixed-length examples with evaluation of coding parameters such as minimum distance?

*) more general discussion on how the unique-neighbor property is useful in coding theory compared to other notions of expansion.

Particular comments:

Before the particulars, a general comment is that some terms that are well-known in the CS-theory community are less known in coding theory. So it is recommended that these terms be defined before they are first used. I give some examples of such in the list below.

1) Page 2 after Theorem 2: shouldn't "degree" change to "average degree" in "with low degree in the induced subgraph"? That is, change to "with low average degree in the induced subgraph".

2) Introduction: some coding-theory motivation should be included here. Can move some from Section 2 and add more from the itemized list above.

3) Page 3 first para of Section 2: rephrase the description of use in Tanner construction. It is not clear why this "goes the other way around"? What is the difference between defining a code C_0 and defining its parity-check graph B_0?

4) Page 3 second+third para of Section 2: expand on coding-theory applications. Currently other applications are emphasized.

5) Page 5 Section 4: introduce "expander mixing lemma" earlier.

6) Page 5 Section 4.1: define \lambda earlier.

7) Page 6 Claim 4: define "Ramanujan graph".

8) same page: define "adjacency operator".

9) Page 7 first para "Interestingly, this result can be deduced almost immediately from Theorem 2": it seems that Corollary 6 gives a little weaker result due to the positive \epsilon.

10) Page 7 Corollary 6: define E_S,V_S.

11) Page 7 Lemma 7: change l to \ell.

12) Page 8: change "sufficiently smalls" to "sufficiently small".

13) Page 8 in definition of operators: define f() and explain its use.

14) Page 9 Lemma 10: define "eigenpair".

15) Page 13 first display equation: explain how this is obtained from (5) in Lemma 10.

16) Page 13 second display equation: missing () in first argument of right-hand side.

17) Page 15 first display equation of Section 6: it seems that (v,j) is not defined.

Author Response

We added some motivation re coding theory, mentioning expander codes and quantum LDPC codes. Our focus was in simplification rather than getting new codes so we didn't for example construct concrete small graphs and the like. We added a discussion about this but it is essentially out of scope for a theoretical paper.

*) what new code parameters are implied by the results? NONE

*) how prior expander-code constructions need to be adapted to use the new graphs. DONE

*) can you show fixed-length examples with evaluation of coding parameters such as minimum distance? no; added a comment

*) more general discussion on how the unique-neighbor property is useful in coding theory compared to other notions of expansion. DONE

Particular comments:

Before the particulars, a general comment is that some terms that are well-known in the CS-theory community are less known in coding theory. So it is recommended that these terms be defined before they are first used. I give some examples of such in the list below.

1) Page 2 after Theorem 2: shouldn't "degree" change to "average degree" in "with low degree in the induced subgraph"? That is, change to "with low average degree in the induced subgraph".

ReWorded to explain: The theorem bounds the average degree of the right-side in the induced subgraph, so there are some right-side verices with low degree. The implication is [S is small] => [N(S) has low avg degree] => [N(S) has some low degree vertices].

2) Introduction: some coding-theory motivation should be included here. Can move some from Section 2 and add more from the itemized list above.

DONE

3) Page 3 first para of Section 2: rephrase the description of use in Tanner construction. It is not clear why this "goes the other way around"? What is the difference between defining a code C_0 and defining its parity-check graph B_0?

Rephrased to make it clear that our construction takes the graph theory approach, and it's only similar to Tanner construction, not implied by it (nor by any code theory results).

4) Page 3 second+third para of Section 2: expand on coding-theory applications. Currently other applications are emphasized.

DONE

5) Page 5 Section 4: introduce "expander mixing lemma" earlier.

DONE

6) Page 5 Section 4.1: define \lambda earlier.

DONE

7) Page 6 Claim 4: define "Ramanujan graph".

Added a reminder about the definition of a bipartite Ramanujan graph above the Claim

8) same page: define "adjacency operator".

DONE

9) Page 7 first para "Interestingly, this result can be deduced almost immediately from Theorem 2": it seems that Corollary 6 gives a little weaker result due to the positive \epsilon.

Thanks for pointing it out; in fact Kahale's result includes a positive \epsilon too. It has been added.

10) Page 7 Corollary 6: define E_S,V_S. 

DONE

11) Page 7 Lemma 7: change l to \ell.

DONE

12) Page 8: change "sufficiently smalls" to "sufficiently small".

DONE

13) Page 8 in definition of operators: define f() and explain its use.

Rephrased to define it without operators but via direct definition of each entry in the matrix.

14) Page 9 Lemma 10: define "eigenpair".

DONE

15) Page 13 first display equation: explain how this is obtained from (5) in Lemma 10.

DONE

16) Page 13 second display equation: missing () in first argument of right-hand side.

DONE

17) Page 15 first display equation of Section 6: it seems that (v,j) is not defined. 

Not sure I understand. The equation defines the set of edges of a graph, whose right side is R x R_0. v is in R, j is in R_0, so (v, j) is a vertex in the graph's right side. With the verbal explanation after that equation, I feel comfortable leaving it like that, don't you think? Maybe we can add just before that "...right side is the cartesian product R × R0 indexed by (v, j) where v \in R and j \in R_0..."

Reviewer 2 Report

Comments and Suggestions for Authors

The paper constructs (slightly) unbalanced unique neighbor expanders. The parameters of their construction follows from the well-known CRVW construction, which gives the stronger notion of lossless expanders, but is considerably simpler.

The construction, in the spirit of Alon-Capalbo, goes roughly as follows. Given:

1) A (c,d)-biregular Ramanujan expander (of growing size, and we know how to construct those efficiently), and,  

2) a great unique-neighbor  (of constant size, so we can brute-force to find it)

Then, put a suitable copy of (2) at every right side vertex of (1), replacing it. The neighbors of the replaced vertex are identified with the left side of (2), and then we re-route in a natural way. This can also be seen as the parity check matrix bipartite graph of a Tanner code with (1) as the base code, and (2) as the inner code.

To establish correctness, they show that in a Ramanujan bipartite graph, every small set on the left side admits neighbours on the right side with low degree in the induced subgraph. It’s a nice theorem by itself, and goes by counting (long) non-backtracking random walks. In particular, it beats what one gets from the expander mixing lemma, which they need.

The construction is nice and natural, and the fact that it achieves the same unique-neighbor property of CRVW makes it worthy of publishing. Recently, two other papers used essentially the same construction to get the  stronger lossless expansion property. The authors should state that (and if so, state that indeed the current paper came before the others).

A few specific comments:

- Theorem 1 is phrased in a weird way. Since we don’t know how \hat{q} depends on c0 yet, this somewhat cumbersome wording doesn’t give us any more information than simply saying “there exists an infinite family of…”. The table that follows is not helpful, and neither is the sentence that follows, since we generally think of \alpha as constant and c0 as the parameter that grows. What is surprising and confusing is the fact that \hat{q} is not monotone in c0, so which c0 should we take? Maybe at this stage it’s better to write that there exists an infinite family, or “for every \alpha there exists q0=q0(\alpha) such that for any q>q0…” and fix c0 to be some arbitrary constant. 

- Theorem 1 and elsewhere - the notion of polynomial time

- Page 2: What do you mean by “low degree vertices”? The number of neighbors? It’s not clear. And also the paragraph that start with “Combining” is not clear. Maybe it’s better to just give the construction itself (at a high level).

- The “Related Work” section contains applications that can be solved also using standard spectral bipartite expanders (and can generally use polishing). Also, the CRVW construction is not such a “different approach” since it also involves constant-sized good objects (yet it is much more involved).

- Beginning of Section 3.2 - why are you only considering d > c, and not d >= c?

- Page 5 - “one-sided spectral graphs” -> “one-sided spectral gap”?

- Page 8 - the fact that “star” = O(1) doesn’t imply that it’s a most 1+eps (which is still true)

- Page 8 - I don’t think that the operator A^LL is defined appropriately (in the summation formula)

- In “Future work” - beyond [AK19], there are recent developments on fast generation of “explicit constructions” under hardness assumptions. It would be good to address those and note what’s practical.

Author Response

- Theorem 1 is phrased in a weird way. Since we don’t know how \hat{q} depends on c0 yet, this somewhat cumbersome wording doesn’t give us any more information than simply saying “there exists an infinite family of…”. The table that follows is not helpful, and neither is the sentence that follows, since we generally think of \alpha as constant and c0 as the parameter that grows. What is surprising and confusing is the fact that \hat{q} is not monotone in c0, so which c0 should we take? Maybe at this stage it’s better to write that there exists an infinite family, or “for every \alpha there exists q0=q0(\alpha) such that for any q>q0…” and fix c0 to be some arbitrary constant. 

We feel that the current wording says for all q> hat q_0, and this is stronger than "infinite family".

- Theorem 1 and elsewhere - the notion of polynomial time

fixed

- Page 2: What do you mean by “low degree vertices”? The number of neighbors? It’s not clear. And also the paragraph that start with “Combining” is not clear. Maybe it’s better to just give the construction itself (at a high level). 

Rewrote

- The “Related Work” section contains applications that can be solved also using standard spectral bipartite expanders (and can generally use polishing). Also, the CRVW construction is not such a “different approach” since it also involves constant-sized good objects (yet it is much more involved).

Right. We added a sentence mentioning that at a high level these are the same, but conceptually "randomness conductors" are more involved. (you need to know much more in the theory of pseudorandomness to understand what's going on)

- Beginning of Section 3.2 - why are you only considering d > c, and not d >= c? 

I think all the claims remain correct for d=c, but it's a bit confusing and we only deal with the d > c case (both the infinite family of expanders, and the small gadget).  

- Page 5 - “one-sided spectral graphs” -> “one-sided spectral gap”? 

Fixed

- Page 8 - the fact that “star” = O(1) doesn’t imply that it’s a most 1+eps (which is still true) 

Correct, we actually meant 1 + o(1). Fixed

- Page 8 - I don’t think that the operator A^LL is defined appropriately (in the summation formula) 

While I think the definition is technically correct, it certainly doesn't help anyone (Reviewer 1 also complained...). What we really want from those operators is how they relate to the number of non-backtracking walks (top of page 13, first sentence in the proof of lemma 8), and the fact that the recursive formulae hold (bottom of page 8). So we revised. 

- In “Future work” - beyond [AK19], there are recent developments on fast generation of “explicit constructions” under hardness assumptions. It would be good to address those and note what’s practical.

We couldn't find something that seemed relevant. Could you possibly provide a pointer?

Round 2

Reviewer 2 Report

Comments and Suggestions for Authors

Recommend to accept after the changes.